# Evaluation of Hard and Soft Tissue Responses to Four Different Generation Bioresorbable Materials-Poly-l-Lactic Acid (PLLA), Poly-l-Lactic Acid/Polyglycolic Acid (PLLA/PGA), Uncalcined/Unsintered Hydroxyapatite/Poly-l-Lactic Acid (u-HA/PLLA) and Uncalcined/Unsintered Hydroxyapatite/Poly-l-Lactic Acid/Polyglycolic Acid (u-HA/PLLA/PGA) in Maxillofacial Surgery: An In-Vivo Animal Study

**DOI:** 10.3390/ma16237379

**Published:** 2023-11-27

**Authors:** Kentaro Ayasaka, Mrunalini Ramanathan, Ngo Xuan Huy, Ankhtsetseg Shijirbold, Tatsuo Okui, Hiroto Tatsumi, Tatsuhito Kotani, Yukiho Shimamura, Reon Morioka, Takahiro Kanno

**Affiliations:** 1Department of Oral and Maxillofacial Surgery, Shimane University Faculty of Medicine, Izumo 693-8501, Shimane, Japan; ayasakakentaro@med.shimane-u.ac.jp (K.A.); m219432@med.shimane-u.ac.jp (M.R.); m239403@med.shimane-u.ac.jp (A.S.); tokui@med.shimane-u.ac.jp (T.O.); tatsumi@med.shimane-u.ac.jp (H.T.); kota0192@med.shimane-u.ac.jp (T.K.); shima06@med.shimane-u.ac.jp (Y.S.); moriokareon@med.shimane-u.ac.jp (R.M.); 2Department of Maxillofacial Surgery, Thong Nhat Hospital, Ho Chi Minh City 700000, Vietnam; ngoxuanhuy158@gmail.com

**Keywords:** osteosynthesis device, bioactive osteoconductivity, bioresorbable material, maxillofacial surgery, bone regeneration

## Abstract

Bone stabilization using osteosynthesis devices is essential in maxillofacial surgery. Owing to numerous disadvantages, bioresorbable materials are preferred over titanium for osteofixation in certain procedures. The biomaterials used for osteosynthesis in maxillofacial surgery have been subdivided into four generations. No study has compared the tissue responses generated by four generations of biomaterials and the feasibility of using these biomaterials in different maxillofacial surgeries. We conducted an in vivo animal study to evaluate host tissue response to four generations of implanted biomaterial sheets, namely, PLLA, PLLA/PGA, u-HA/PLLA, and u-HA/PLLA/PGA. New bone volume and pertinent biomarkers for bone regeneration, such as Runx2, osteocalcin (OCN), and the inflammatory marker CD68, were analyzed, and the expression of each biomarker was correlated with soft tissues outside the biomaterial and toward the host bone at the end of week 2 and week 10. The use of first-generation biomaterials for maxillofacial osteosynthesis is not advantageous over the use of other updated biomaterials. Second-generation biomaterials degrade faster and can be potentially used in non-stress regions, such as the midface. Third and fourth-generation biomaterials possess bioactive/osteoconductivity improved strength. Application of third-generation biomaterials can be considered panfacially. Fourth-generation biomaterials can be worth considering applying at midface due to the shorter degradation period.

## 1. Introduction

The maxillofacial region is unique and is one of the most important dynamic regions of the body. The influence of saliva, along with a vast community of microorganisms, a strong occlusal load, and muscular forces, poses a challenging setting for wound healing and the quicker resumption of function. Osteosynthesis devices are most commonly used in the maxillofacial region to achieve the anatomic restoration of fractured bony segments, bridge segmental bone defects after tumor resection, and in dentofacial deformity correction surgery to secure the position of the bone with stability after osteotomy [1]. The first metallic plate system for rigid fixation was described by Carl Hansmann and Sir William Lane approximately one century ago [2]. Titanium plates and screws, for many decades, have been considered the gold standard osteosynthesis material, which has two main variants: (1) load-bearing osteosynthesis allows placement of the devices in areas with poor bone stock/poor bone quality, and (2) load-sharing offers adequate rigidity while transferring functional loads to the bone underneath. Load-bearing systems are placed after segmental bone resection in the cases of tumors/cancers. A load-sharing system is used to provide adequate stability in cases of trauma or orthognathic surgery. Titanium devices have potential risks such as meterosensitivity, palpation of plates and screws owing to thickness, production of radiographic artifacts, leaching of titanium particles into host tissues, a stress-shielding effect, the need for a second surgery for removal, and growth restriction when used in pediatric patients [3,4,5,6]. Titanium particles have been detected in human lymph nodes and soft tissues a year after surgical intervention, and traces have been observed in the spleen and associated lymph tissues that provide systemic transmission [6] and intracranial migration after craniofacial surgery [7].

### 1.1. Background of the Study

The search for alternative biomaterials began with bioresorbable systems gaining popularity because of their ability to degrade inside host tissues without producing toxic effects and negating the disadvantages associated with titanium devices. An ideal biomaterial should support the underlying bone, aid regeneration, degrade completely, and be absorbed by the host once healing is complete [8]. It should be strong, flexible, and nontoxic.

The first in vivo study to test biodegradable materials was performed in animals in 1966 [9,10]. Bioresorbable materials are currently classified into four generations based on their mechanical/physicochemical properties and biochemical behavior inside the host tissue. First-generation biomaterials consist of homopolymers, poly-L-lactic acid (PLLA) and poly-D-lactide (PDLA) [11]. Second-generation biomaterials include copolymers, PLLA and polyglycolic acid (PGA) admixture in varied concentrations [12]. Third-generation biomaterials are composites of inorganic/bioceramic fibers/particles and homogenous polymers, uncalcined/unsintered hydroxyapatite (u-HA) and PLLA; and fourth-generation comprises composites of inorganic/bioceramic fibers/particles and copolymers, u-HA, PLLA, and PGA. A short summary of bioresorbable materials from each generation is provided below.

The use of biomaterials for clinical applications began with the introduction of PGA [13]. PGA is highly crystalline in nature with increased molecular weight and shows rapid degeneration [14], with most of its molecular mass being depleted within 4–7 weeks [12]. Intense inflammatory exudates with acid degeneration byproducts are the main adverse effects associated with PGA.

Polylactic acid (PLA) was developed subsequent to PGA as a bioresorbable material. Two stereoisomeric forms of PLA with different properties owing to the optically active carbon in lactic acid exist, i.e., PLLA and PDLA [15]. PLLA has been used since the 1990s as a material for maxillofacial osteosynthesis [11]. The hydrophobic nature of PLLA renders it resistant to hydrolysis; hence, it retains its original strength over a prolonged period. The resorptive time of PLLA is estimated to be about 3.5 years in vivo [16]. PLLA has several disadvantages, such as foreign-body reaction, chronic inflammation [17] owing to late degradation [18], and inadequate material intensity. Conversely, PDLA can lead to giant cell inflammation as host tissues cannot digest D-lactide. Plate palpability is reportedly due to the thickness of the plate in the thin facial regions. A modified thinner plate version of PLLA has also been made available, but its use remains limited to the bones of the midface [19]. However, these biomaterials do not possess adequate rigidity compared to titanium systems.

Copolymers of first-generation biomaterials were created by modifying the lactide and glycolide material ratio to elicit better host responses and decrease inflammation. This type of co-polymerization is known to have better mechanical characteristics and the degradation proceeds by hydrolysis. An equal ratio of 50/50 PLLA/PGA has the fastest resorption time [20]. At a ratio of 85/15 PDLA/PGA, complete resorption is slightly prolonged. Copolymers of PLLA/PDLA at a ratio of 70/30 lose strength 2 days after in vivo placement. Although they can be applied to midfacial osteosynthesis [21], they are limited because they lack bioactive/osteoconductivity.

Newer biomaterial synthesis has been initiated for clinical translation to obtain better clinical outcomes. HA was added to PLLA to induce bioactive/osteoconductivity. The forged composite osteosynthesis materials are composed of u-HA with a weight of 30% for miniscrews and 40% for miniplates [22]. The clinical use of u-HA/PLLA has favorable effects as it allows for complete bone tissue ingrowth and replacement [23]. They have been used with good success and stability in the midface and mandibular osteosyntheses. u-HA/PLLA systems have very good mechanical properties and bond directly to bone but also possess limitations such as a long resorption period of nearly 4–5 years in vivo, coupled with the discomfort caused by fibrous tissue formation [24].

A relatively new material with a modified composition, u-HA/PLLA/PGA, a forged composite with copolymers from the second generation, has been introduced to circumvent the disadvantages of the third-generation material, u-HA/PLLA. Previous studies have revealed that the ability of u-HA/PLLA/PGA is comparable to that of u-HA/PLLA in new bone formation and osteoconductivity, even with a lower amount of u-HA, through periostin-induced bioactive cell accumulation [25] and faster degradation. It also showed a shorter resorption time than u-HA/PLLA, with profoundly less inflammation [26].

Recently, computer-aided designing (CAD) to generate 3-dimensional (3D) printing of polymeric materials is gaining popularity in biomaterial applications, as it enables the production of complex geometric anatomical patterns and scaffold designs to enable bone regeneration. They possess several advantages in being flexible yet having adequate stiffness, biocompatibility, and conductivity [27]. Currently, polyphosphazenes, silicone blocks, polyvinyl chloride, foaming silicones, polyurethane, and siphenyle polymers have gathered attention as materials commonly employed for the fabrication of maxillofacial prosthesis [28]. Polyetheretherketone (PEEK) implants have been successfully applied to zygomatic defects with good success rates [29].

### 1.2. Rationale

Although the salient features of each material, as well as reviews outlining each generation of biomaterials, have been published earlier, no basic comparative research study has analyzed the effects of all four generations of biomaterials currently available in clinical applications worldwide and reported the clinical applicability of each of these materials as osteosynthesis devices in maxillofacial surgery.

### 1.3. Aim of the Study

Keeping these points in mind, we conducted this animal study to evaluate the features of host reactionary responses to biomaterial sheets in vivo—both hard and soft tissues—including the estimation of the ability of the bioresorbable sheets to produce new bone, inflammation assessment, and sheet resorbability using a rat mandibular model. Our study was designed to simulate the role of osteosynthesis devices used for maxillofacial bone fixation and the feasibility of their potential clinical application in the maxillofacial region.

## 2. Materials and Methods

### 2.1. Materials Used

Four sheets, one from each generation of bioresorbable materials, were used in this study (manufactured by Teijin Medical Technologies Co., Ltd., Osaka, Japan). The dimensions of the implanted sheet were 5 mm × 5 mm × 0.3 mm (length × width × thickness) (Figure 1). The sheets were made into square-shape to maintain the uniformity of size and standardization of shape across all groups. The PLLA sheets consisted of 100% pure PLLA. The PLLA/PGA polymers contained a mixture of 85% PLLA and 15% PGA. The u-HA/PLLA and u-HA/PLLA/PGA-forged composite sheets had compositions of 40% u-HA with 60% PLLA and 10% u-HA with 90% PLLA/PGA, respectively (the PLLA/PGA proportion was 88:12). The synthesis of u-HA was performed by the hydrolysis of pure calcium hydrogen phosphate anhydride and calcium carbonate heated to 90 degrees; the particles were then filtered, incubated, and dried. This process generated large u-HA particles that were then crushed and sieved. The mean particle size of the u-HA in both u-HA/PLLA and u-HA/PLLA/PGA was 3–5 μm [30]; the calcium-phosphorous molar ratio was 1.69. The mechanical properties and tensile strengths of various osteosynthesis systems have been well-iterated in the literature [1,19]. Since the bioresorbable material sheets used in this study were fabricated for the purpose of conducting this research, we did not determine the tensile strength of the sheets.

### 2.2. Animal Protocol: Surgical Procedure and Sacrifice

The animal experimental protocol was approved by the Animal Ethics Committee of Shimane University (Approval number: IZ3-110). The Guidelines for Care and Use of Laboratory Animals, Shimane University Faculty of Medicine, Izumo, Shimane, were also observed. Twenty-six Sprague–Dawley male rats (Charles River, Tokyo, Japan) aged 10 weeks with an average weight of 300 g were allotted into study and sham control groups according to our previous research methodology. The rats were subdivided into two study groups. Group A received a PLLA sheet on the right side of the mandible and a PLLA/PGA sheet on the left side of the mandible. Group B received the u-HA/PLLA sheet on the right side of the mandible and the u-HA/PLLA/PGA sheet on the left side of the mandible. One rat was assigned to the sham control group at each time point (Figure 2). The rats in the sham group did not receive any material. The allotment of animals into the respective study groups and the number of specimens analyzed are described in detail in Figure 3.

The surgical procedure was performed under sterile, aseptic conditions. All rats received general anesthesia using a three-mixture solution of midazolam (2 mg/kg), medetomidine hydrochloride (0.15 mg/kg), and butorphanol (2.5 mg/kg) intraperitoneally as per our previously established protocol. After skin disinfection, a full-thickness skin incision of approximately 1 cm in length was made in the submandibular region. The subcutaneous, muscular, and periosteal layers were incised to expose the mandibular angle region. The buccal aspect of the bone was covered with the respective material sheet, and a hemoclip was fixed to maintain the vertical position of the material sheet in proximity to the host bone (Figure 4). Placement of the hemoclip prevents the dislodgement of the material sheet into the surrounding soft tissues. The soft tissue wound was closed in layers using resorbable sutures. Antibiotics (Enrofloxacin 1 mL/kg) and analgesics (Buprenorphine 0.25 mL/kg) were administered subcutaneously postoperatively. The rats regained consciousness approximately 1.5–2 h post anesthesia and exhibited normal appetite and behavior.

At 2 and 10 weeks after surgery, the rats were sacrificed through an inhalation anesthetic overdose as per the protocol. Because the hemi-mandibles contained sheets of different materials, the mandible was cut at the midline, and each side was collected separately. Specimens were stored and preserved in 10% neutral-buffered formalin for microcomputed tomography (micro-CT) and immunohistochemistry (IHC) analyses. Additionally, three material sheets from each study group (total = 12) were collected at each point in time for molecular weight calculations to ascertain the degradation time in vivo.

### 2.3. Assessment of New Bone Formation Using Micro-CT

The extracted rat hemi-mandibular specimens were scanned using a high-resolution 3D Micro X-ray CT CosmoScan FX scanner (Rigaku Corporation, Tokyo, Japan). The imaging conditions used to scan the specimens were as follows: voltage, 90 kV; current, 88 μA; duration, 2 min; FOV, 10.24 × 10.24 × 10.24 mm; matrix, 512 × 512 × 512; and resolution, 20 × 20 × 20 μm. Bone mineral density phantoms of 0, 50, 200, 800, and 1200 mg HA/cm^3^ were employed. Every harvested specimen was scanned, excluding those from which the material sheets were extracted at each time point for molecular weight calculation.

The scanned images of each specimen were analyzed using the ImageJ software (Fiji, version 2.14.0/1.54f) to evaluate the amount of new bone growth outside the material sheet. The two-dimensional (2D) images opened using ImageJ were initially displayed in the axial view. We used a specific rotation plugin, TransformJ, to enable the viewing of 2D images in the sagittal plane (Figure 5A). In the sagittal view, the cross-sections clearly indicated the material sheet location, host bone, new bone outside the sheet, and hemoclip (Figure 5B). Only the third-generation material sheet, u-HA/PLLA, was visually discernible in the micro-CT section because of the higher concentration of u-HA. We manually traced the new bone on the outer side of the sheet in each slice using the ‘Polygonal selection tool.’ The traced region was saved with the ‘Region of Interest (ROI) Manager’ tool. After measurement of the entire specimen, the individual areas marked were measured using the ‘Measure’ option in the ROI manager. The cumulative area was then calculated (Figure 5C). The total new bone volume was estimated with the formula V = ∑Si × d, where V is the volume in mm^3^; Si, the new bone area in a single micro-CT slice in mm^2^; and d, the thickness of one micro-CT slice (0.02 mm) [25].

### 2.4. Tissue Conditioning for IHC Evaluation

The scanned specimens were demineralized with 10% ethylenediaminetetraacetic acid for 4–5 weeks. The specimens were then embedded in paraffin and made into blocks using a sealed automatic fixation and embedding device and carefully sectioned at a thickness of 4 μm. The slide preparation for the IHC evaluation was carried out by Sept. Sapie Co., Ltd. (Tokyo, Japan).

### 2.5. Hematoxylin-Eosin (HE) and IHC Staining

Sliced paraffin-embedded sections were stained with HE for histological assessment. For IHC staining, the sections were deparaffinized with xylene and rehydrated with ethanol. Antigen activation treatment was performed, and 3% hydrogen peroxidase was applied to block endogenous peroxidases. For primary staining, the slides were treated with an anti-CD68 rabbit polyclonal antibody (Abcam: ab125212; dilution 1:1000), mouse anti-human osteocalcin monoclonal antibody (BIO-RAD: 0400-0041; dilution 1:3000), and anti- runt-related transcription factor x type 2 (Runx2) rabbit polyclonal antibody (Abcam: ab23981; dilution 1:300) at room temperature for 50 min. Subsequently, the samples were washed with phosphate-buffered saline (PBS) (0.01 M, pH 7.4) three times for 5 min each. Secondary staining was performed using the Histofine Simple Stain Rat MAX-PO (MULTI) (Nichirei Bioscience Inc., 414191) at room temperature for 30 min. The slides were again washed with PBS and treated with diaminobenzidine (DAB) coloring for 10 min. Cellular nuclei were stained with Meyer’s hematoxylin for 30 s for anti-CD68 slides and for 2 min for anti-Runx2 and osteocalcin slides. After a final wash under running water to remove the color, the slides were dehydrated, permeabilized, and sealed for microscopic visualization. We did not perform IHC staining for the sham control group.

### 2.6. IHC Assessments

The stained IHC slides were visualized using a BX43 light microscope (Olympus Corporation, Tokyo, Japan), and images were captured at different magnifications for observation and further analysis. Three specimens from each group were used for the IHC evaluation. We used the ImageJ software for all IHC analyses. CD68 and Runx2 expression were measured using a cell labeling index. Three images were captured from each slide at 40× magnification, both outside and inside the sheet, in proximity to the host bone. The “Cell Counter” tool was used to estimate the positive and negative cells by manual marking. The percentage of positive cells was calculated using the following formula: percentage of positive cells = no. of positive cells/no. of positive cells + no. of negative cells × 100. The average score of all three specimens was used to obtain the final score. OCN antibodies accumulate in the extracellular matrix; hence, we used DAB staining intensity for estimation. The areas outside the material sheet and the host bone close to the sheet were analyzed. Three specimens from each group were used to compare the OCN expression. With regard to DAB, a lower score equates to a darker signal and increased staining intensity. We computed the digital H-scores for OCN expression using the “Color Deconvolution” tool in ImageJ. The intensity range was 0–255. The region of OCN deposition was marked manually using the ROI Manager, and a measurement tool was used to obtain the area intensity value. This value was subtracted from 255 to obtain the digital H-score [31] (Figure 6).

Three individual researchers performed the IHC data analysis, and the average values were represented to avoid observer-related data bias.

### 2.7. Estimation of Bioresorbable Sheet Resorption In Vivo

The molecular weights of the sheets at weeks 2 and 10 were compared with those of the unused sheets to infer material degradation. The assessment was performed using gel chromatography. An HLC-8320GPC instrument (TOSOH Corporation, Tokyo, Japan) was used. Tetrahydrofuran was used as the mobile phase at a flow rate of 0.35 mL/min. The following formulas were employed: (retention rate) R = Mw2/Mw0 and R = Mw10/Mw0, where Mw2, Mw10, and Mw0 are the molecular weights of the material sheets at the end of weeks 2 and 10, respectively, and Mw0 is the control.

### 2.8. Statistical Analysis

SPSS for Mac OS version 27.0 (IBM Corporation, Armonk, NY, USA) was used for the statistical analysis. Non-parametric Kruskal–Wallis and Mann–Whitney U tests were used to compare significant differences between the study groups at weeks 2 and 10. The Wilcoxon signed-rank test was used to compare the significance of the same study group between the time points. Statistical significance was set at *p* ≤ 0.05. We used the Bonferroni correction to reduce the probability of errors (*p* ≤ 0.016) due to multiple comparisons.

## 3. Results

### 3.1. Volume of New Bone Formed Outside Sheet

New bone formation was visually discernible in all four groups at each time point (Figure 7). The u-HA/PLLA/PGA sheets are not visible on the radiographs because of their reduced HA content (10%). The volume of newly formed bone increased between the two-time points in all groups. By week 2, the PLLA group showed negligible new bone formation on the outer surface of the sheet. The PLLA/PGA group showed slightly more bone formation than the PLLA group. The u-HA/PLLA and u-HA/PLLA/PGA groups showed comparable amounts of bone formation at week 2. By the end of week 10, PLLA had the lowest volume of new bone compared to PLLA/PGA. Interestingly, the u-HA/PLLA/PGA group showed a greater new bone volume than the u-HA/PLLA group at week 10. No changes were observed in the sham group at either time point.

The mean new bone volume generated in the PLLA and PLLA/PGA groups by the end of week 2 were 0.14 mm^3^ and 0.32 mm^3^, respectively. Bone volume in the u-HA/PLLA and u-HA/PLLA/PGA groups were 0.58 mm^3^ and 0.60 mm^3^, respectively. There was a significant difference in new bone volume between the PLLA and u-HA/PLLA/PGA groups at week 2. At week 10, the mean bone volume was calculated to be 0.63 mm^3^ for PLLA and 0.93 mm^3^ for PLLA/PGA, which was almost similar between the two groups but had no statistical significance. However, the new bone formation at different time points was significant for both PLLA and PLLA/PGA groups. u-HA/PLLA/PGA had the highest new bone volume (2.69 mm^3^) among all groups by week 10, surpassing u-HA/PLLA (1.64 mm^3^), but this difference was insignificant. Both u-HA/PLLA and u-HA/PLLA/PGA showed significant increases in bone volume compared with PLLA and PLLA/PGA by week 10 (Figure 8).

### 3.2. HE Staining

In week 2, PLLA displayed a thick layer of fibrous tissue surrounding the biomaterial sheets. Small stumps of bony extensions from the host bone were observed at the top and bottom of the sheet. Minimal osteoblastic activity was observed at the ends of the sheets. However, the outer surface of the sheet showed only a fibrous layer and no osteoblasts. PLLA/PGA showed a similar histological presentation. The host bone-sheet interface had considerable osteoblastic activity. No osteoblastic cells were observed on the outer surface of the sheet. The histological features of the u-HA/PLLA and u-HA/PLLA/PGA were identical. A thick layer of osteoblastic cells lined the host-bone-sheet contact region. The entire span of the sheet in contact with the host bone contained a swarm of osteogenic cells. The outer portion of the sheet had an immature woven bone attached directly to the sheet surface. Osteoblastic cells surrounded the newly formed bone outside the sheet. Angiogenesis was evident in both groups during the initial stage. In both the u-HA/PLLA and u-HA/PLLA/PGA groups, fibrous tissue was seen adjacent to the sheets at week 2, but their thicknesses were less than those observed in the PLLA and PLLA/PGA groups. New bones in the u-HA/PLLA and u-HA/PLLA/PGA were also observed to attach to and extend from the host bone around the lower part of the sheet to the outside of the sheet, as shown in Figure 9.

At week 10, the PLLA group still had a relatively thick fibrous tissue layer, suggesting a chronic inflammatory reaction, while the fibrous layer surrounding the PLLA/PGA group had a slight decrease in diameter. No osteoblastic cells were observed in either group. In both groups, the sheet was in direct contact with the host bone. A very small amount of new bone was attached to the outer surface of the sheets, and the bone volumes of PLLA and PLLA/PGA were comparable. The u-HA/PLLA and u-HA/PLLA/PGA groups had solid, mature, and lamellar bone on their outer surfaces, anchored to the host bone from the lower end of the material sheets. The newly formed bone was firmly attached to the outer part of the sheet. Several osteocytes were trapped in the lacunae of the newly formed bone. A few osteoblasts were observed enclosing the area where the bone was still being deposited on the outer surface of the sheet. A thin layer of osteoblastic cells lining the host bone was also observed. The fibrous tissue thickness reduced considerably faster in the u-HA/PLLA/PGA group than in the u-HA/PLLA group. The sham group remained the same at both time points, with no significant differences (Figure 9).

### 3.3. IHC Analysis

#### 3.3.1. Runx2 Expression

In week 2, several layers of positive cells surrounding the newly formed bone in u-HA/PLLA and u-HA/PLLA/PGA groups accumulated on the outer part of the material sheets, with a slightly higher response to u-HA/PLLA/PGA. Only the tips of the bony stumps at the ends of the sheets toward the outer part showed positive staining for Runx2-cells in PLLA. PLLA/PGA had scattered Runx-2 positive cells on the outer aspect of the sheet, and their expression was noted to be lower than that in the PLLA group. In week 10, Runx2 expression decreased considerably across all groups. However, in the u-HA/PLLA and u-HA/PLLA/PGA groups, although a decrease was observed, the expression was higher than that in the PLLA and PLLA/PGA groups (Figure 10A).

In week 2, Runx2-positive cells were observed lining the host bone sheet surface in the PLLA and PLLA/PGA groups. The expression was more intense and vivid in the host bone regions in contact with the u-HA/PLLA and u-HA/PLLA/PGA; it was observed to be a highly interactive region filled with osteogenic cells. At a later time point (by week 10), a decrease in the levels of antibody-stained cells in all groups was noted (Figure 10B).

u-HA/PLLA and u-HA/PLLA/PGA showed statistically significant results for Runx2 expression in the outer sheet region compared to PLLA and PLLA/PGA at both time points. Runx2 expression in PLLA was significantly higher than in PLLA/PGA at week 2 (Figure 11A). Regarding the host bone-sheet surface, PLLA and PLLA/PGA exhibited higher levels of antibody-positive cells than the outer sheet region, but the expression was not as marked as in the u-HA/PLLA and u-HA/PLLA/PGA groups. The PLLA group showed statistically significant differences between different time points near the host bone-sheet interface. The u-HA/PLLA and u-HA/PLLA/PGA groups showed a significant increase in Runx2 labeling in the host bone lining sheet compared to all other groups and at different time points within their respective groups (Figure 11B).

#### 3.3.2. OCN Expression

Week 2 OCN expression was predictably low in all the groups because of immature bone formation. However, the host bone that was immediately in contact with the biomaterial sheet showed a positive antibody response. Bony stumps extending from the host bone were observed around the lower end of the sheet. By week 10, the PLLA and PLLA/PGA groups exhibited decreased staining on the outer aspect, signifying a lower volume of new bone. The u-HA/PLLA and u-HA/PLLA/PGA groups exhibited a marked increase in OCN antibodies on the outer sheet surface. In the u-HA/PLLA group, the new bone that formed on the outer side of the sheet was independent, whereas in the u-HA/PLLA/PGA group, it extended from the host bone (Figure 12).

The expression of OCN in the outer sheet region in the PLLA group was greater than that in the PLLA/PGA group at weeks 2 and 10, but the difference was insignificant. There were no significant differences between the different time points in any group. u-HA/PLLA exhibited the highest OCN expression at weeks 2 and 10; the difference was statistically significant between u-HA/PLLA and PLLA, as well as PLLA/PGA, but not with u-HA/PLLA/PGA (Figure 13A). Similar to the host bone side, PLLA/PGA had more OCN expression than PLLA by week 2. The H-score by week 10 in both groups was increased and comparable, with no statistical difference. u-HA/PLLA showed a significantly higher expression at Week 2 on the host bone side than PLLA/PGA. The u-HA/PLLA/PGA group had the highest score by Week 10, with no significant difference compared to the other groups; however, the u-HA/PLLA group displayed marked significance compared to PLLA and PLLA/PGA at week 10 (Figure 13B).

#### 3.3.3. CD68 Expression

In week 2, numerous clusters of cells and individual cells in layers were scattered on the outer surfaces of the sheets in the PLLA and PLLA/PGA groups. This concentration was significantly higher in the PLLA group. Although it contained a considerable amount of PLLA, the u-HA/PLLA group had only a few layers of scattered cells on the outer sheet region, whereas CD68 positive cells lined the outer side of the u-HA/PLLA/PGA sheet as a thin layer. In all the groups, antibody-positive cells were present within the confines of the fibrous tissue around the sheet. In week 10, PLLA still showed several layers of CD68-positive cells, suggesting the progression of an active inflammatory process. The PLLA/PGA had fewer cell layers. The percentage of antibody-positive cells decreased by week 10 in u-HA/PLLA and u-HA/PLLA/PGA (Figure 14A).

By week 2, the CD68 marker close to the host bone showed very little response in the u-HA/PLLA and u-HA/PLLA/PGA groups. In other words, the positive cells toned down compared to the soft tissue response observed outside the sheet. PLLA showed clusters of cells lodged at the host bone-sheet interface. PLLA/PGA also showed layers of CD68 cells by week 2. By week 10, all groups showed a reduction in the number of CD68 cells, including the PLLA and PLLA/PGA groups (Figure 14B).

PLLA and PLLA/PGA showed the highest percentage of positive cells on the outer sheet region by week 2, with no statistically significant difference at week 10. The percentage of positive cells outside the sheet was higher in u-HA/PLLA than in u-HA/PLLA/PGA; however, by week 10, there was a reduction in u-HA/PLLA. The cell-labeling index of u-HA/PLLA/PGA did not decrease until week 10 (Figure 15A). In week 2, near the host bone, the PLLA group had the highest significance score, followed by the PLLA/PGA group. Interestingly, the percentage did not decrease much by week 10 in either the PLLA or PLLA/PGA groups and remained significantly higher than that in the other two groups. The u-HA/PLLA and u-HA/PLLA/PGA groups exhibited significantly lower positive cell scores at weeks 2 and 10; however, there was no significant difference between weeks 2 and 10 (Figure 15B).

### 3.4. Bioresorbable Sheet Resorption Rate

The homogeneous and copolymer materials underwent degradation at varying but predictable rates (Figure 16). The PLLA, u-HA/PLLA, and u-HA/PLLA/PGA maintained almost their original weights at the end of week 2, with averages of 91.8%, 86%, and 86.13%, respectively. PLLA/PGA degraded faster, with an average score of 63.6% at week 2. u-HA/PLLA/PGA degraded more slowly than plain PLLA/PGA owing to its u-HA content. By the end of week 10, the PLLA (52.4%) and u-HA/PLLA (55.1%) groups retained more than half of their weight. Between weeks 2 and 10, PLLA/PGA resorption (39.8%) was slow and steady. The molecular weight of the u-HA/PLLA/PGA changed drastically, with an average of 40.1%. Though both second and fourth-generation materials contain PGA, the fourth generation contains a lesser amount of PGA (12%) than compared to the second generation (15%). The fourth generation is also composed of u-HA, which could possibly contribute to the delay in degradation, as u-HA is eliminated by phagocytosis. The difference in resorption demonstrated by each group between the two time points and between different groups at the same time point was statistically significant.

## 4. Discussion

Recently, bioresorbable materials have been favored over conventional cumbersome metallic osteosyntheses, and numerous attempts to replace them have been made across various specialties. These materials are superior to metallic implants in terms of biocompatibility, lack of secondary surgery, absence of stress shielding [1], no adverse effects on bone growth, bioactive/osteoconductivity (third and fourth generations), inability to produce radiographic distortion, no mutagenic effects [15], and alteration of material components to obtain better in vivo responses [32]. The materials are now termed ‘bio-resorbable’ to enunciate the osteoconduction brought forth in vivo [17]. Although these biomaterials have been studied separately under different in vitro and in vivo settings by numerous researchers, they have not been studied together. We conducted this study to specifically analyze host reactions to these biomaterials from hard and soft tissue viewpoints and quote their usage specificity and usefulness as maxillofacial osteosynthesis materials.

### 4.1. Biomaterial Induced Bone Formation

Micro-CT sections were used to calculate the volume of new bone formed on the outer regions of the sheets. Micro-CT evaluation aids in the assessment of bone volume in three dimensions and allows accurate estimation. PLLA showed minimal new bone formation; the PLLA/PGA group had more new bone formation than the PLLA group by week 2. In the PLLA group, an acidic pH owing to chronic inflammation caused by the release of small degradation particles can prevent bone deposition and mineralization, which may also result in the resorption of the newly formed bone [33]. Reactionary bone formation may also occur in the PLLA group, wherein inflammatory conditions can stimulate the host tissue to deposit bone [34]. Although PLLA/PGA produces fewer inflammatory reactions than PLLA, the lack of bioactivity causes less attraction of osteogenic cells and, therefore, less bone formation. The amount of new bone in the PLLA and PLLA/PGA groups increased in volume by week 10 but was significantly lower than that of the materials containing u-HA. The u-HA/PLLA and u-HA/PLLA/PGA groups exhibited greater new bone volumes owing to their u-HA content. Although the u-HA content was lower in u-HA/PLLA/PGA (10%) than in u-HA/PLLA (40%), more new bone was formed in u-HA/PLLA/PGA than in u-HA/PLLA. These findings are consistent with previous literature [26]. The amount of new bone formed in the u-HA/PLLA/PGA group was significantly higher than that in all other groups by Week 10. The u-HA/PLLA group exhibited comparable amounts of new bone. The new bone in u-HA/PLLA/PGA extended from the host bone; however, in u-HA/PLLA, the bone attached to the sheet was independent of host bone contact. This feature of bone formation with host bone contact was observed earlier when u-HA/PLLA meshes were used in vivo to avoid parent bone contact [35]. Bone quality in both groups was homogenous. The bioactive/osteoconductive tendency exhibited by u-HA in third and fourth-generation materials is because of a layer of calcium phosphate (CaP) that leaches out of the material and the concomitant fibrous tissue that is deposited around the biomaterial [32] following the initiation of degradation. The CaP layer eventually attracts and reacts with the serum protein fibronectin. This forms a bed for the attachment of osteoblastic progenitor cells. Fibronectin then attaches to the cell transmembrane protein, integrin. Molecular interactions between the formed layer and consequent signaling mechanisms induce osteoblast differentiation and matrix mineralization [36], thus forming new bone. It has been noted that proteins from fluids surrounding biomaterials are involved in ionic exchange. Serum proteins are adsorbed by calcium phosphate when dissolution of the biomaterial occurs. The interaction and adsorption of these factors are largely dependent on factors such as crystallinity, texture, and composition of the biomaterial. When the said proteins adsorb onto the calcium phosphate substrate, their charge, availability of free calcium, and their concentration determine further surface-coverage. These proteins then influence the formation of new calcium phosphate crystals over time. The quality, nature, and conformation of these proteins that interact on the biomaterial surface then control the cellular activity [37]. The expression of the osteogenic cell biomarker Runx2 and matrix protein OCN mirrored the increase in bone volume in the u-HA groups. This remarkable bioactive/osteoconductivity feature is absent in PLLA and PLLA/PGA. The increased bone volume observed in the u-HA/PLLA/PGA with a lower u-HA content was owing to the faster degradation setting [38] of PLLA/PGA (88:12 ratio) present in the material. The degradation rates of u-HA/PLLA and u-HA/PLLA/PGA in week 2 were comparable. After week 2, the rapid degeneration of the copolymer PLLA/PGA caused the release of more u-HA particles at a faster rate, thus explaining our results. In contrast, in u-HA/PLLA, hydrolysis-resistant PLLA prevented the rapid release of u-HA.

### 4.2. IHC Analysis—Runx2, OCN Biomarker Relevance

Relevant biomarker expression and timed identification form the basis of evidence-based studies that attempt to prove regeneration. We evaluated two biomarkers, Runx2 and OCN, to assess bone regeneration. Runx2 activation causes an increase in other markers, such as OCN, collagen type I, and alkaline phosphatase [39]. Runx2 is a hallmark bone regeneration marker, as it is vital for osteoblast transcription and maturation [40] and is the most prevalent protein in osteoblastic cells [41]. The involvement of Runx2 in early development, particularly in intramembranous ossification and tooth formation, is well-known. Runx2 expression is also regulated by signaling and various factors such as bone morphogenic protein, estrogen, vitamin D_3_, and transforming growth factor-β [42]. In our study, we observed the high expression of Runx2 in cells in week 2, lined up at the host-material interface and outer surface of the sheet adjacent to newly formed bone relative to u-HA/PLLA and u-HA/PLLA/PGA materials, thus demonstrating their bioactive behavior. The PLLA group showed a slightly higher expression of Runx2 than the PLLA/PGA group. The expression gradually decreased by week 10, with a smaller number of cells observed around the PLLA and PLLA/PGA sheets. The acidic pH and inflammatory environment around PLLA can stimulate new bone formation and attract osteogenic cells [34]. PLLA/PGA is devoid of bioactive/osteoconductivity and hence did not show many antibody-positive cells surrounding the sheet. The host bone reaction to u-HA/PLLA and u-HA/PLLA/PGA was marked in terms of Runx2 positive antibody cells. As Runx2 accumulates in areas where bone formation is active, its expression was higher at the initial stages and then tapered gradually by week 10. OCN is a non-collagenous bone matrix protein. OCN transcription is dependent on vitamin D [43]. Several physiological endocrine functions have been identified for OCN [44]. The carboxylated form of OCN binds freely to calcium ions and hydroxyapatite, exhibiting its role in bone mineralization and promoting better bone quality and quantity [36]. A mature, carboxylate form of OCN is present intracellularly for secretion into the bone matrix. OCN is primarily known to bind to the calcium in HA when released into the bone micro-environment by undergoing morphological change. Through the tight complex formed between OCN-HA and collagen, OCN aids in bridging the matrix and mineral components of bony tissue [45]. The OCN expression scores in PLLA and PLLA/PGA were much lower at Week 2 but showed improvement by Week 10. The dominant expression of OCN in the u-HA/PLLA group, followed by the u-HA/PLLA/PGA group, was observed by week 10 outside the sheet owing to early maturation and mineralization. The u-HA/PLLA/PGA group exhibited the highest intensity of OCN with respect to the host bone lining of the material sheets.

### 4.3. Inflammatory Conditions Elicited around the Biomaterials—CD68 Analysis

We analyzed the expression of CD68 as a reliable surface marker of macrophages [46]. In our study, the expression of CD68 increased in the outer sheet region in the PLLA and PLLA/PGA groups. The intensity was more pronounced near the host interface across all groups. The expression of CD68 relative to that of u-HA/PLLA and u-HA/PLLA/PGA decreased and reached a minimum by week 10; however, the expression was consistent in both PLLA and PLLA/PGA groups.

Every implanted biomaterial induces an inflammatory response in the host tissues, the effects of which can either be beneficial in bringing about subsequent host tissue regeneration or provoke deleterious conditions in the host body [47]. This response is mainly mediated via innate immunity [48] through M1 macrophages, which secrete inflammatory mediators, and subsequently through M2 macrophages, which elicit healing responses [49]. Acquired immune responses also play a role by inducing pro and anti-inflammatory reactions [50]. Normal cell-mediated immune responses cause the replacement of the biomaterial with host tissue, while disproportionate responses lead to chronic host reactions [47]. Previous studies have suggested two main hypotheses to explain foreign body reactions: (i) the formation of a semi-permeable fibrous capsule that prevents the passage of degraded polymer particles, thereby resulting in increased osmotic pressure, and (ii) acidic polymer particles that evoke inflammation by rendering the pH acidic and possibly cause dissolution of the regenerated bone [51]. Crystalline particles that cannot be ingested by phagocytes remain inside the foreign body giant cells, forming a hydrophobic complex nidus for prolonged periods of time [52,53]. Understanding these responses is vital when dealing with biomaterial placement in facial bones, especially the mandible. This is because the mandible has reduced vascularity and is less resistant to these effects owing to increased occlusal load. T-cell recruitment is a mandatory process that enables bioactivity. Activated T-cells gather and surround the material and release CCL5, which attracts mesenchymal stem cells (MSCs). MSCs can differentiate into precursors, osteoblasts, and bone deposits. Chemotactic migration of fibroblast cells and MSC has been observed under decreased inflammatory conditions [54]. Zhao et al. showed that techniques employing minimally invasive surgery cause reduced inflammation and thereby hinder regeneration owing to reduced inflammation. Therefore, inadequate inflammation is associated with reduced immune cell recruitment and less osteoconductivity [55].

PLLA fragments are quite potent in invoking a strong reaction, and these byproducts have been identified in the body approximately 5 years post-placement [18]. Palpable inflammatory masses formed in relation to the periorbital fixation of PLLA and PLDLA at various time points and their excision has been reported. The excised tissues show intense inflammatory reactions with giant cells, fibroblasts, lymphocytes, and histiocytes [56]. PLLA/PGA systems have reportedly resolved completely without causing late inflammatory reactions [57]. In pediatric patients who underwent craniofacial fixation with PLLA/PGA plates and screws, existing inflammatory conditions were transient and resolved once degradation of the copolymer was finished (>9 months) [58]. This correlates with our findings of increased inflammation during weeks 2 and 10. u-HA/PLLA screws show minimal CD68 markers peripherally around the material [1]. The u-HA/PLLA underwent stable hydrolysis, with most of the inflammation subsiding after 52 weeks [59]. There is evidence that u-HA particles come in contact with the bony surface, enabling bone bonding without fibrous tissues [60]. This suggests better biocompatibility with host tissues and applicability to a variety of clinical scenarios [61], suggesting that the addition of u-HA tends to lower the inflammatory activity of PLLA [30]. Thinner biomaterials exhibit a better host tolerance [62,63]. A dominant population of CD68 + cells can represent lymphocytic cell populations (T-cells), suggesting that healing could occur simultaneously at later stages [46].

### 4.4. Bioresorbable Sheet Resorption Characteristics

Biomaterial degradation is conducive to tissue growth and normal function [64]. In this study, PLLA and u-HA/PLLA exhibited similar resorption rates, whereas PLLA/PGA and u-HA/PLLA/PGA resorbed faster. Degradation of these materials follows a particular pattern: at first, there is a loss of molecular weight, followed by a reduction in strength reduction, and finally, a decrease in mass. The u-HA particles must be phagocytosed by giant cells/macrophages. PLLA undergoes hydrolytic degradation initially and enzymatic degradation later [65]. This process begins at the periphery, which is the region most exposed to water, and later progresses toward the center. The center of the material quickly degrades as the lactic acid concentration increases owing to central accumulation and cannot be released freely from the material surface [32]. The u-HA particles are of an organic ceramic nature and have a structure that closely resembles that of native bone. PLLA is nontoxic and breaks down into l-lactic acid, which, when compared to the d-lactide produced by PDLA, has a lower chance of initiating chronic inflammation marked by foreign body giant cell reactions [66]. PLLA/PGA resorption initiates through the depolymerization of ester bonds by hydrolysis, followed by consumption and breakdown into carbon dioxide and water [67]. Because of their amorphous structure, PLLA/PGA materials do not release crystals and show a steady degradation rate with increased host tolerance [12]. They have a resorption time of 12–18 months and are engineered to provide support for bone regeneration over 6–8 weeks [57]. u-HA/PLLA exhibits a bending strength equivalent to that of human cortical bone for 25 weeks in vivo. It is evident that PLLA disappears 4 years later, and most of the material is replaced by bone 5.5 years after placement in vivo [24]. Bone formation and replacement in and around the biomaterial occur simultaneously, and bone is deposited where the biomaterial is absorbed [68]. u-HA/PLLA/PGA has a resorption time similar to that of plain PLLA/PGA. It maintains adequate strength during initial healing and degrades faster, thus allowing bone replacement between osteosynthesis components. The above-mentioned are findings from our previous research works that have been correlated with other existing literature.

### 4.5. Application of Each Generation Bioresorbable Material in Maxillofacial Osteosynthesis

#### 4.5.1. Bioresorbable Osteosynthesis in Orthognathic Surgery

Unlike titanium, bioresorbable systems prevent the need for a second surgery for removal [69]. LeFort I osteotomies can be stabilized with customized bioresorbable Y-plates for the nasomaxillary buttress and L-shaped plates for the zygomaticomaxillary buttress, which is also feasible for multi-segment maxillary osteotomies. With Bi Sagittal Split Ramus Osteotomy (BSSRO), two resorbable miniplates are usually recommended, one superior and the other inferior to the mandibular canal. If only one miniplate is placed across the mandibular segments after BSSRO, stability can be compromised [70]. The PLLA/PGA devices used for maxillary osteosynthesis show favorable stability and low complication rates [71]. u-HA/PLLA devices provide osteoconductivity and bone-binding abilities, and u-HA/PLLA meshes have been used to promote stability after BSSRO [72]. Reliable stability was observed after bimaxillary orthognathic surgery and plating using the bioresorbable systems [73]. In their review article, Park et al. stated that no patient in their experience of 13 years with second-generation materials had fragment displacement after bimaxillary surgery, even with the judicious use of elastics to guide occlusion [17]. Two bicortical self-reinforcing-PLLA screws (20 mm in length) used for BSSRO (mandibular advancement) in 11 patients provided stable results and good outcomes [74]. The u-HA/PLLA showed good stability and could be used for the midfacial, maxillary, and mandibular segments. Fast resorption is a concern when utilizing u-HA/PLLA/PGA systems in the mandible.

#### 4.5.2. Bioresorbable Osteosynthesis in Maxillofacial Trauma

The clinical application of bioresorbable materials for maxillofacial fracture fixation was first reported in 1971 [75]. Resorbable systems were introduced in 1987 to fix zygomatic bone fractures [76]. PLLA/PGA material under the commercial names Lactosorb^®^ (Biomet Inc., Jacksonville, FL, USA) and RapidSorb^®^ (DePuy Synthes CMF, West Chester, PA, USA) are highly used as midface and maxillary fixation devices. Pediatric mandibular fractures are commonly fixed using resorbable materials (Second generation), and it has been shown that in dentate segment fractures with simultaneous condylar fractures, temporomandibular joint ankylosis can be prevented by resorbable plate fixation in the dentate region and early joint mobilization [77]. Plain PLLA and u-HA/PLLA plates and screws are comparable to titanium in terms of stability and outcomes when used at the midface [22]. Orbital floor fractures can also be managed using u-HA/PLLA sheets with favorable outcomes [8]. Plate palpability is reduced in patients undergoing resorbable osteosynthesis [78]. Many authors do not recommend the use of first and second-generation materials for mandibular fractures because of inadvertent muscle pulling [79]. Currently, there is no concrete evidence to indicate that biodegradable materials can exhibit load-bearing properties [80]; however, their feasibility is being evaluated by our team.

Based on the characteristics of the bioresorbable materials analyzed in our study, it is evident that these biomaterials can possibly be useful as osteosynthesis devices in maxillofacial surgery. The resistance of PLLA to hydrolysis indicates its strength during the initial healing phase. PLLA/PGA degraded faster, retained adequate strength in the initial phase, and did not retain any residual material. Both u-HA/PLLA and u-HA/PLLA/PGA exhibited bioactive/osteoconductivity and attracted an increased number of osteogenic cells and their progenitors to the vicinity of the biomaterials. In our study, both materials showed increased bone formation and active recruitment of osteogenic cells. The combination of PLLA strengthens u-HA/PLLA, as slow degradation initially improves the mechanical strength and u-HA release over an extended period, thus resulting in improved bioactivity. Faster degradation of u-HA/PLLA/PGA allows for the early replacement of bony tissues instead of osteosynthesis. Combining polymers and copolymers permits the exploitation of the beneficial outcomes of both materials while creating the desired clinical outcomes.

### 4.6. Limitations of Our Study

One disadvantage of bioresorbable systems is their inability to be discerned on radiographs. u-HA/PLLA is the only material that is visible because of the high amount of HA, which can be modified to enable radiographic visibility. Our study has the following limitations: (1) the number of rats was small due to the replacement, reduction, refinement (RRR) principle, and (2) specimen analyses were performed at only two time points. We hope to overcome the shortcomings in our future research.

## 5. Conclusions

The main goals of maxillofacial surgical procedures include rapid bony union and quick postsurgical functional resumption. To summarize the feasibility of their application as maxillofacial osteosynthesis devices, early materials such as PGA and PLLA have limited applications as fixation devices and can be considered feasible for use only in regions of reduced tension and muscular activity. PLLA/PGA has the efficient property of faster degradation and can be applicable for fixation of the midfacial and periorbital regions. Newer materials, such as u-HA/PLLA and u-HA/PLLA/PGA, have superior characteristics, such as rigidity and bioactivity/osteoconductivity, which are comparable to those of titanium devices. u-HA/PLLA is worthy of application to the midface, maxilla, and mandible, including the mandibular condyle. The use of u-HA/PLLA/PGA in midfacial bones may be considered more reliable; however, the resorption time is quicker; hence, stability during the healing period may be a key concern. Accordingly, clinicians should assess each patient’s specific situation and decide which material would be beneficial in providing the most favorable outcome.

## Figures and Tables

**Figure 1 materials-16-07379-f001:**
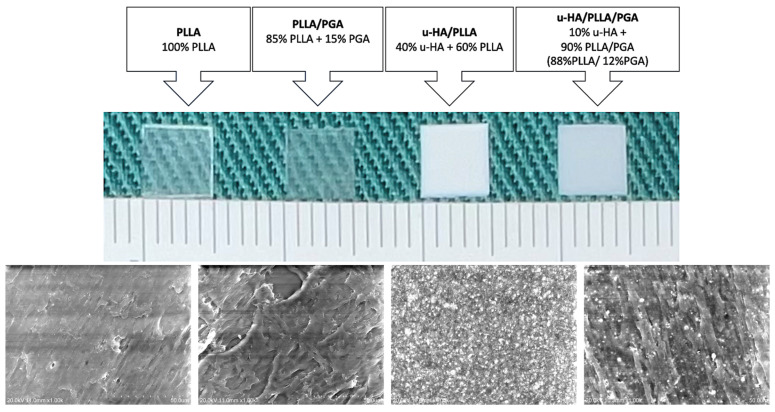
Implanted material sheets. From left to right: PLLA, PLA/PGA, u-HA/PLLA, u-HA/PLLA/PGA with their respective compositions. Below—SEM images ×1000 magnification (from left to right): PLLA, PLLA/PGA, u-HA/PLLA, u-HA/PLLA/PGA. PLLA, poly-L-lactic acid; PGA, polyglycolic acid; u-HA, uncalcined/unsintered hydroxyapatite; SEM, scanning electron microscope.

**Figure 2 materials-16-07379-f002:**
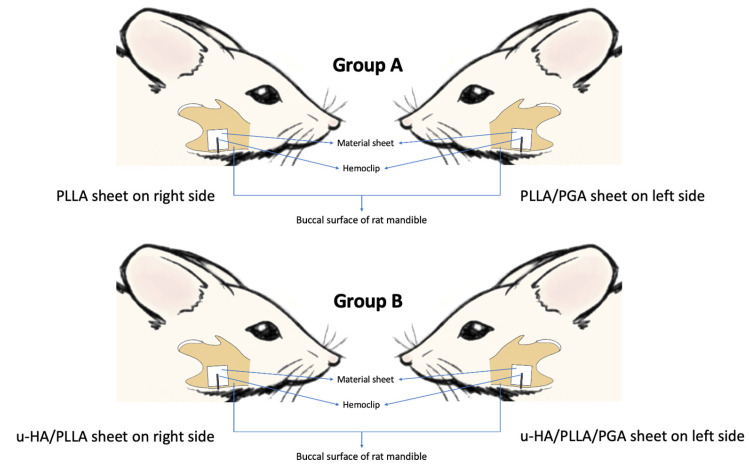
Animated image showing the position of bioresorbable material sheets onto the rat mandible. PLLA, poly-L-lactic acid; PGA, polyglycolic acid; u-HA, uncalcined/unsintered hydroxyapatite.

**Figure 3 materials-16-07379-f003:**
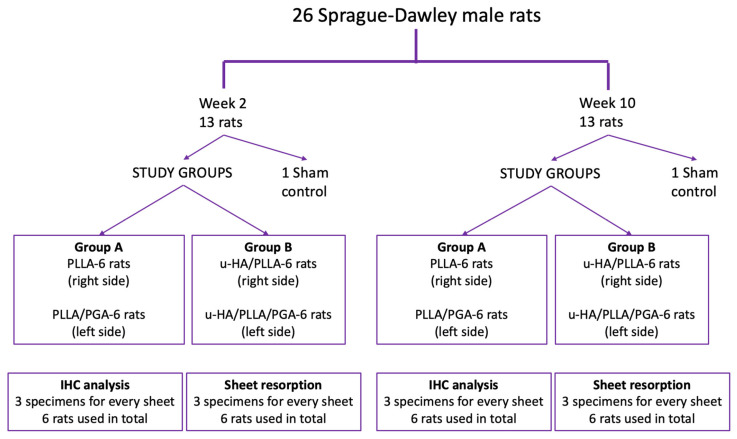
Flowchart demonstrating allotment of animals into respective groups. PLLA, poly-L-lactic acid; PGA, polyglycolic acid; u-HA, uncalcined/unsintered hydroxyapatite; IHC, immunohistochemical analysis.

**Figure 4 materials-16-07379-f004:**
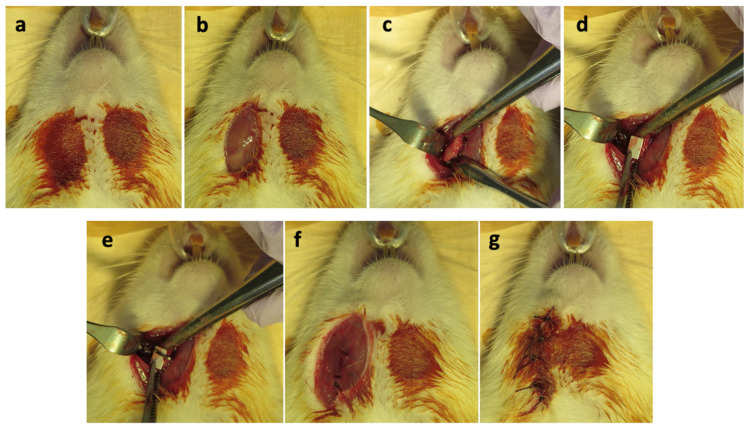
Surgical procedure: (**a**) skin disinfection, (**b**) skin incision at the submandibular region, (**c**) exposure of mandible, (**d**) placement of material sheet, (**e**) fixation with hemoclip, (**f**) wound closure in layers, (**g**) skin closure and completion.

**Figure 5 materials-16-07379-f005:**
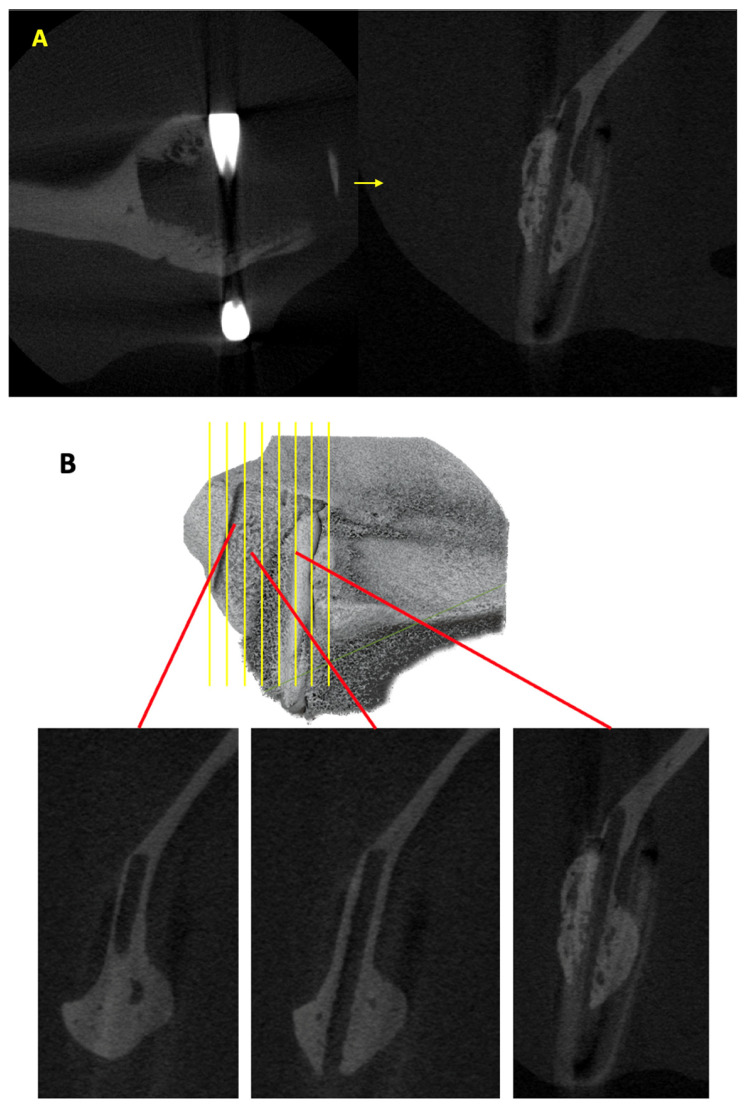
(**A**) Rotation of the section using the ImageJ plugin ‘TransformJ’ to view the image in the sagittal plane. Host bone, material relation to host bone, and newly formed bone can be differentiated clearly when viewed from the sagittal plane. (**B**) Sagittal images show newly formed bone on the outer aspect of the material sheet from the anterior to the posterior region in ImageJ. (**C**) Tracing the area of new bone from each slide using a polygonal selection tool and measuring the new bone area using ROI Manager in ImageJ.

**Figure 6 materials-16-07379-f006:**
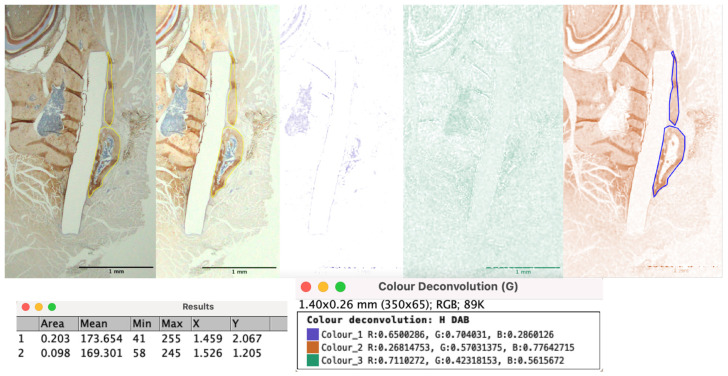
OCN intensity measurement using DAB color deconvolution and results from ROI marking in ImageJ. DAB, diaminobenzidine; OCN, osteocalcin; ROI, region of interest.

**Figure 7 materials-16-07379-f007:**
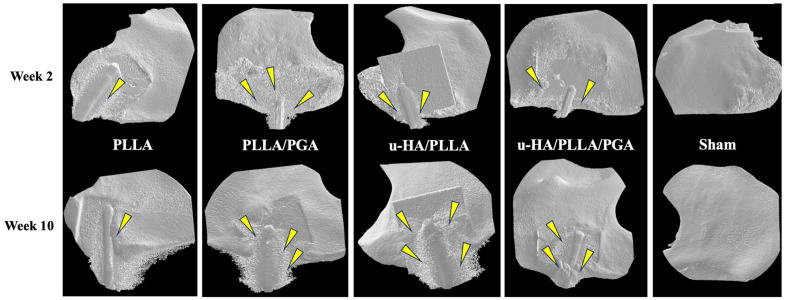
Three-dimensional images obtained from micro-CT evaluation. Note that only the u-HA/PLLA/PGA sheet is visible among the four generations owing to higher HA content. Yellow arrows indicate areas of new bone formation. PLLA, poly-L-lactic acid; PGA, polyglycolic acid; u-HA, uncalcined/unsintered hydroxyapatite.

**Figure 8 materials-16-07379-f008:**
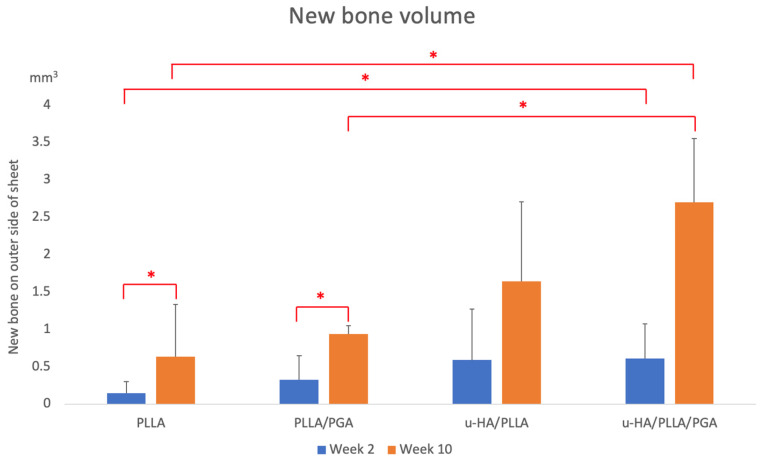
Results of new bone formed on the outer side of the sheet in each group. * Indicates statistical significance. PLLA, Poly-L-lactic acid; PGA, polyglycolic acid; u-HA, uncalcined/unsintered Hydroxyapatite.

**Figure 9 materials-16-07379-f009:**
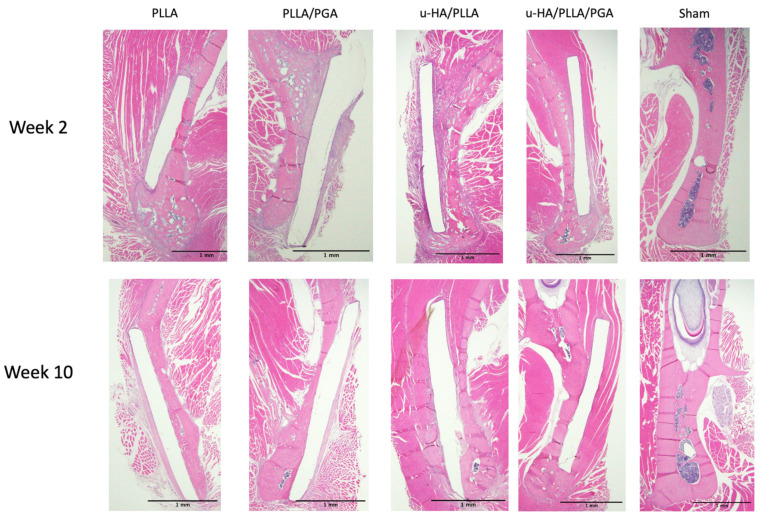
HE staining of specimens by the end of weeks 2 and 10. Inflammation and fibrous tissue formation surrounding biomaterial sheets were higher in the PLLA and PLLA/PGA groups (×1.25 magnification, Scalebar = 1 mm). PLLA, poly-L-lactic acid; PGA, polyglycolic acid; u-HA, uncalcined/unsintered hydroxyapatite; HE, hematoxylin and eosin.

**Figure 10 materials-16-07379-f010:**
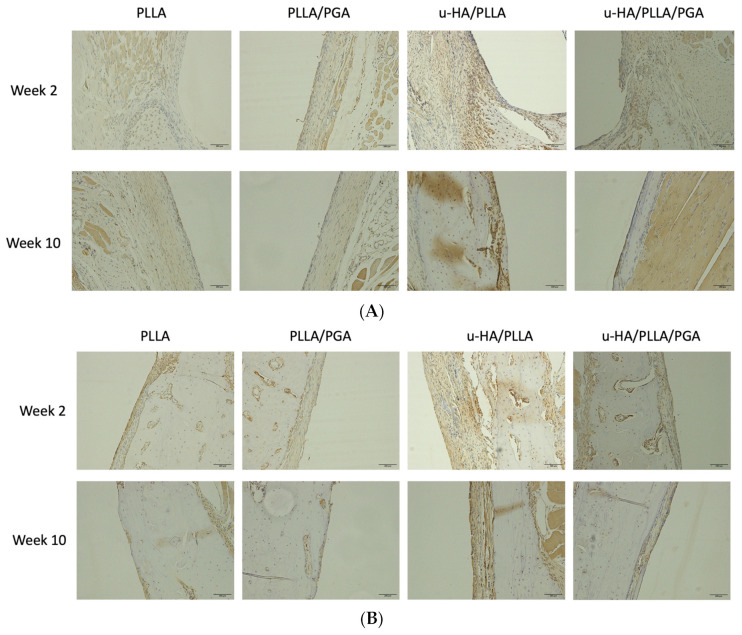
(**A**) Runx2 expression as seen on the outer region of the sheet. Bony stumps from the bottom of the sheet showing Runx2 expression in the PLLA group by week 2 (×20 magnification, Scalebar = 100 μm). PLLA, poly-L-lactic acid; PGA, polyglycolic acid; u-HA, uncalcined/unsintered hydroxyapatite; Runx2, runt-related transcription factor x type 2. (**B**) Runx2 expression near host bone. Several layers of bioactive cells were visible by week 2 in groups containing u-HA as opposed to scanty cell expression in PLLA and PLLA/PGA (×20 magnification, Scalebar = 100 μm). PLLA, poly-L-lactic acid; PGA, polyglycolic acid; u-HA, uncalcined/unsintered hydroxyapatite; Runx2, runt-related transcription factor x type 2.

**Figure 11 materials-16-07379-f011:**
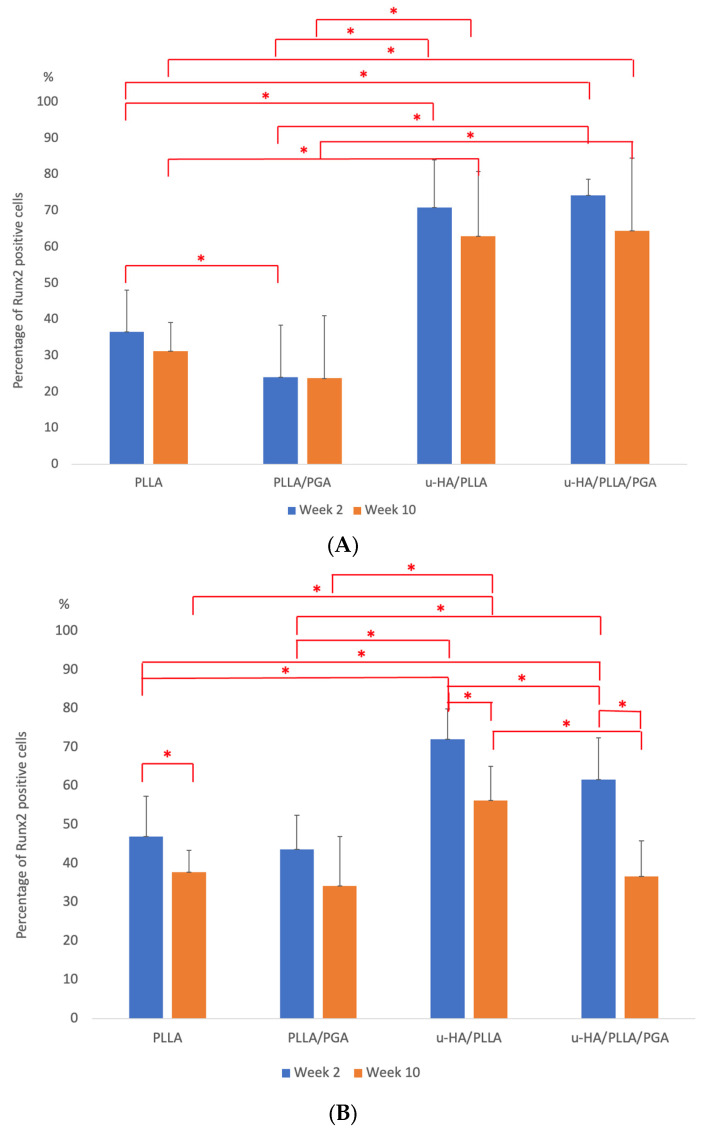
(**A**) Graph showing cell labeling results of Runx2-cell accumulation in the outer sheet region. * Indicates statistical significance. PLLA, poly-L-lactic acid; PGA, polyglycolic acid; u-HA, uncalcined/unsintered hydroxyapatite; Runx2, runt-related transcription factor x type 2. (**B**) Graph showing cell labeling results of Runx2-cell accumulation near host bone. * Indicates statistical significance. PLLA, poly-L-lactic acid; PGA, polyglycolic acid; u-HA, uncalcined/unsintered hydroxyapatite; Runx2, runt-related transcription factor x type 2.

**Figure 12 materials-16-07379-f012:**
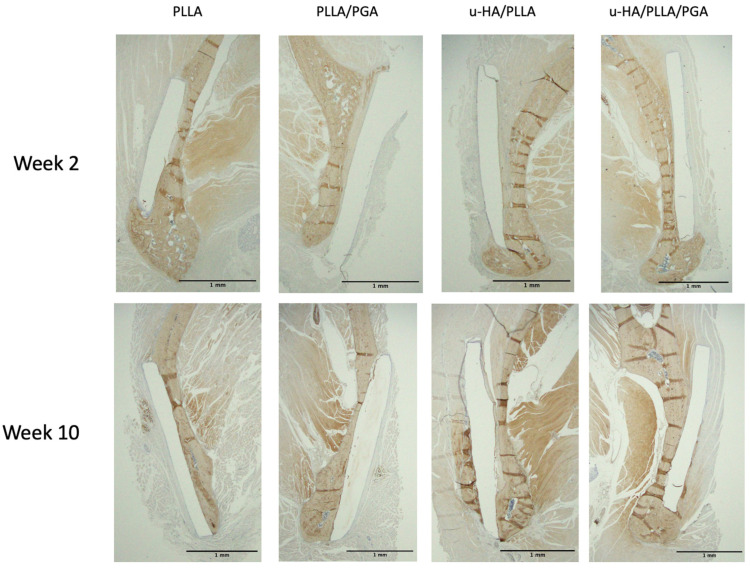
OCN expression surrounding biomaterial sheets at weeks 2 and 10. OCN intensity was marked by week 10 in the u-HA/PLLA and u-HA/PLLA/PGA groups (×1.25 magnification, Scalebar = 1 mm). PLLA, poly-L-lactic acid; PGA, polyglycolic acid; u-HA, uncalcined/unsintered hydroxyapatite; OCN, osteocalcin.

**Figure 13 materials-16-07379-f013:**
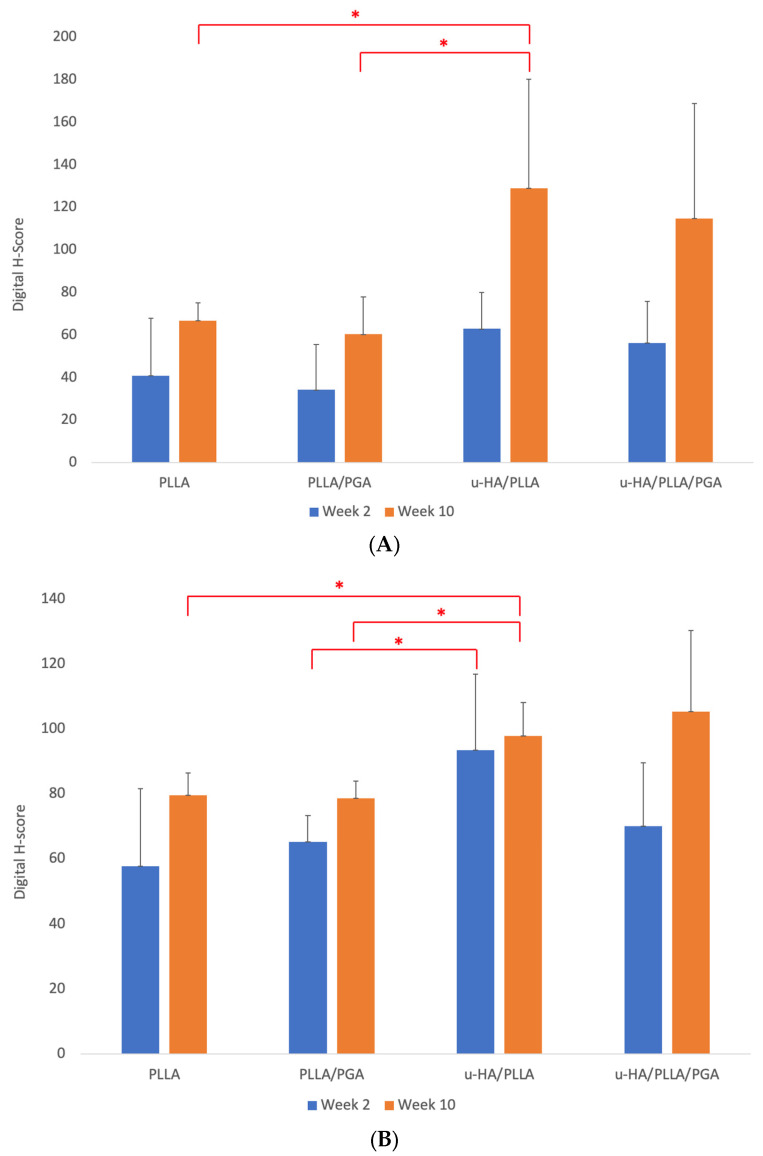
(**A**) Graph showing OCN accumulation outside sheet. * Indicates statistical significance. OCN, osteocalcin. (**B**) OCN accumulation adjacent to host bone. * Indicates statistical significance. OCN, osteocalcin.

**Figure 14 materials-16-07379-f014:**
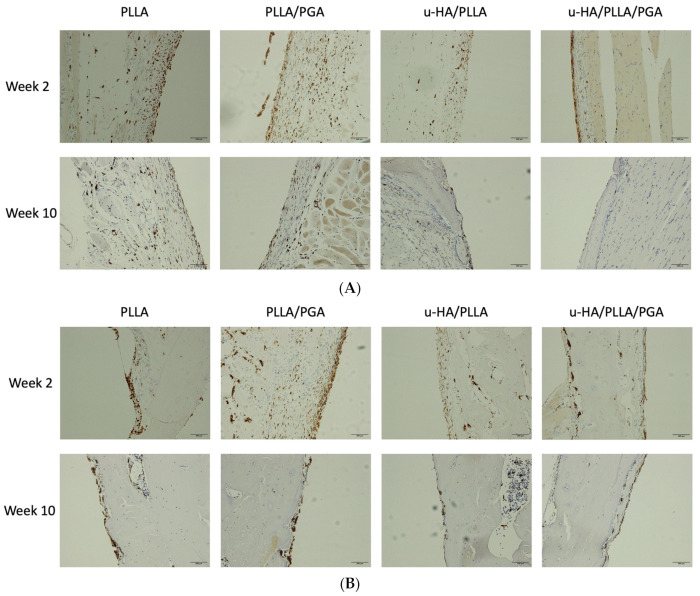
(**A**) CD68 cells on the outer region of the implanted material sheet. Week 2 inflammatory profile was higher in PLLA and PLLA/PGA. The CD68-positive cells were found attached to the fibrous tissue encapsulation (×20 magnification, Scalebar = 100 μm). PLLA, poly-L-lactic acid; PGA, polyglycolic acid; u-HA, uncalcined/unsintered hydroxyapatite. (**B**) CD68 cells closer to the host bone. Significant inflammation was still visible in the PLLA and PLLA/PGA groups. There was a drastic decrease in CD68 cell expression by week 10 in the PLLA/PGA group (×20 magnification, Scalebar = 100 μm). PLLA, poly-L-lactic acid; PGA, polyglycolic acid; u-HA, uncalcined/unsintered hydroxyapatite.

**Figure 15 materials-16-07379-f015:**
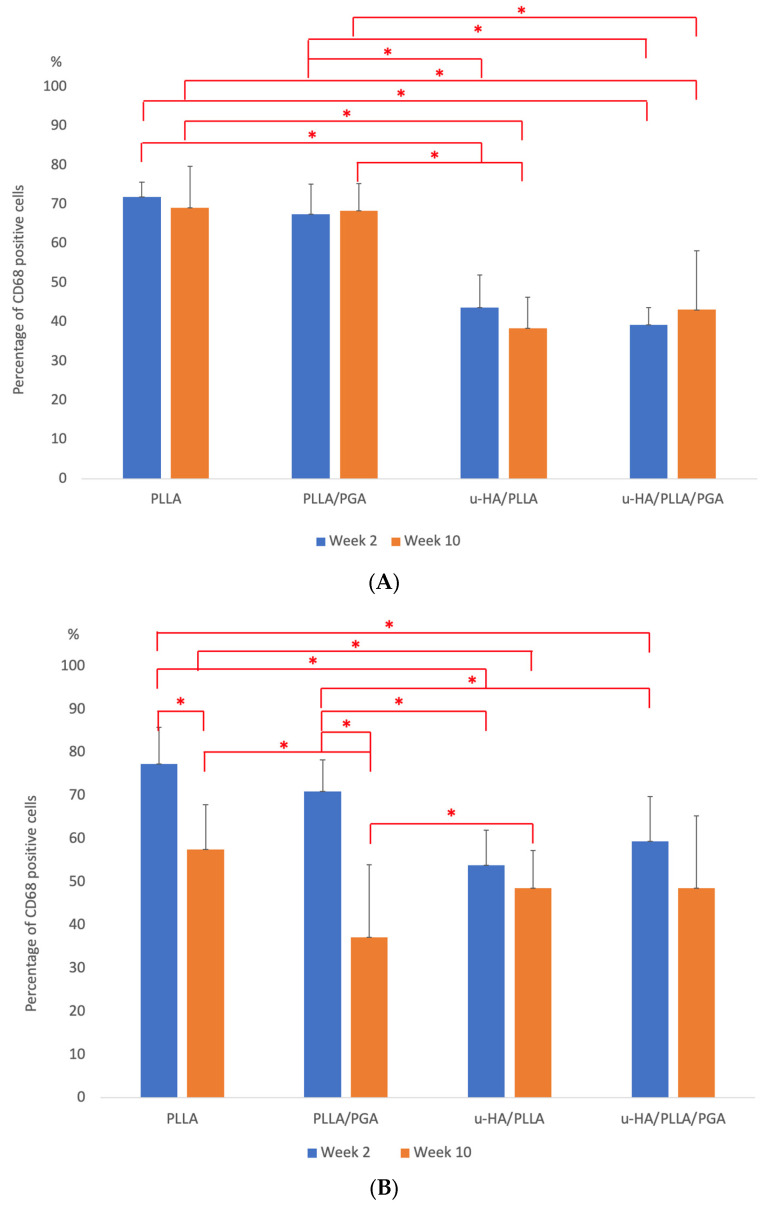
(**A**) CD68 expression outside sheet. * Indicates statistical significance. PLLA, poly-L-lactic acid; PGA, polyglycolic acid; u-HA, uncalcined/unsintered hydroxyapatite. (**B**) CD68 expression close to host bone. * Indicates statistical significance. PLLA, poly-L-lactic acid; PGA, polyglycolic acid; u-HA, uncalcined/unsintered hydroxyapatite.

**Figure 16 materials-16-07379-f016:**
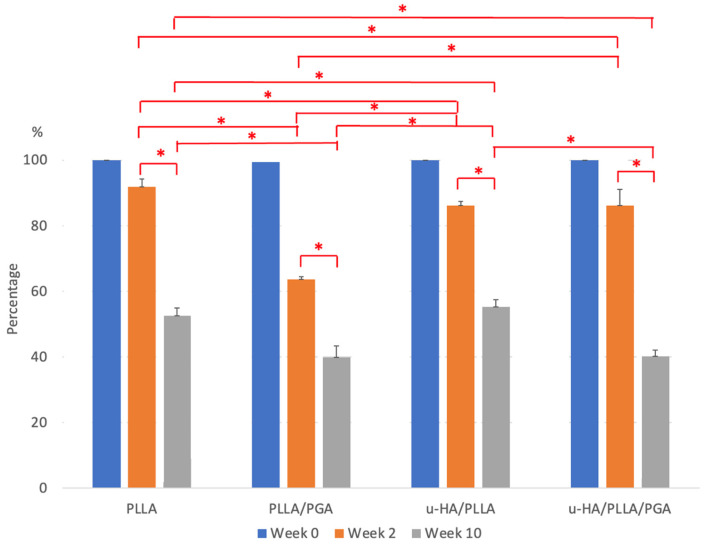
Percentage of bioresorbable material retained inside the host body at ascribed time points of Weeks 2 and 10. * Indicates statistical significance. PLLA, poly-L-lactic acid; PGA, polyglycolic acid; u-HA, uncalcined/unsintered hydroxyapatite.

## Data Availability

All data have been illustrated in the manuscript.

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
