# Peer review of "(untitled)"

_materials, 2023, doi:10.3390/ma16237379_

Round 1

Reviewer 1 Report

Comments and Suggestions for Authors

This paper conducted an animal study to evaluate the features of host reactionary responses to biomaterial sheets in vivo—both hard and soft tissues—including the estimation of the ability of the bioresorbable sheets to produce new bone, inflammation assessment, and sheet resorbability using a rat mandibular model. 

This paper is novel and the experimental design is reasonable. It is recommended to accept the paper directly after improving the coherence of the Abstract and Introduction.

Author Response

Comments 1: This paper conducted an animal study to evaluate the features of host reactionary responses to biomaterial sheets in vivo—both hard and soft tissues—including the estimation of the ability of the bioresorbable sheets to produce new bone, inflammation assessment, and sheet resorbability using a rat mandibular model. 

This paper is novel and the experimental design is reasonable. It is recommended to accept the paper directly after improving the coherence of the Abstract and Introduction.

Response 1: We thank the reviewer for their kind comments. We have made changes to improve the clarity of the Abstract and Introduction as per your suggestion as well as through incorporation of the suggestions of other reviewers.

Reviewer 2 Report

Comments and Suggestions for Authors

The article's authors presented a study evaluating the soft and hard tissue response to four generations of bioresorbable materials for craniofacial surgery. In-vivo studies were conducted on animals. 

The authors' study has important implications regarding reconstructing damaged bony areas of the craniofacial region. The article was written carefully in terms of editing and content. I have minor comments and questions about the content of the article. 

a) I would ask you to include the numerical values obtained in the research process in the abstract.

b) I would ask in the introduction to include information about the applicability of 3D printing in manufacturing models of anatomical structures, surgical templates, and implants of the facial area. Recently, we have seen a significant increase in the applicability of polymeric materials.

c) Have the authors conducted tests on the materials received, for example, Rockwell hardness, Charpy Impact Strength, or determination of static tensile strength?

d) Why didn't the authors use a semi-automatic segmentation method in the process of assessing bone volume? e.g. the most well-known one, i.e. thresholding? The process of manually generating a polygonal curve is very long and does not always involve an accurate determination of the volume of the object. 

Comments on the Quality of English Language

The paper is reasonably properly written (but the article needs a typo-grammatical check).

Author Response

Comments 1: The article's authors presented a study evaluating the soft and hard tissue response to four generations of bioresorbable materials for craniofacial surgery. In-vivo studies were conducted on animals. 

The authors' study has important implications regarding reconstructing damaged bony areas of the craniofacial region. The article was written carefully in terms of editing and content. I have minor comments and questions about the content of the article. 

a) I would ask you to include the numerical values obtained in the research process in the abstract.

Response 1: Thank you for the above comment. We acknowledge the reviewer’s suggestion in including the numerical values to the Abstract section. However, we were unable to do so and apologize for the same, considering the fact that our study comprises of the evaluation of individual responses of four generation bioresorbable materials and not their comparison. The analyses have been performed at two regions, i.e., outside sheet and host bone. There is also a 200-word restriction in the Abstract section, and it is difficult to incorporate the scoring/ percentages of all the biomarkers analyzed.

Comments 2: b) I would ask in the introduction to include information about the applicability of 3D printing in manufacturing models of anatomical structures, surgical templates, and implants of the facial area. Recently, we have seen a significant increase in the applicability of polymeric materials.

Response 2: Thank you for the suggestion. We have added a short paragraph containing information on Computer-Aided Designing (CAD) and 3-Dimensional (3D) printing of polymeric materials applied to the facial region in the form of Patient Specific Implants (PSI), Maxillofacial Prosthesis along with their features and advantages in Line 118 along with references 27, 28, 29.

Comments 3: c) Have the authors conducted tests on the materials received, for example, Rockwell hardness, Charpy Impact Strength, or determination of static tensile strength?

Response 3: Thank you for the above query. The tensile strength of the bioresorbable plates and screws made from PLLA, PLLA/PGA, u-HA/PLLA, and u-HA/PLLA/PGA that are commercially available for clinical use have been evaluated and reported previously in literature (references quoted in manuscript text). The material sheets used in this study were specially fabricated for the purpose of studying host tissue response only and is not suitable for clinical application. Hence, we did not determine tensile strength of the material sheets. This explanation been added in the Materials & Methods section under subsection Materials used – Line 156.

Comments 4: d) Why didn't the authors use a semi-automatic segmentation method in the process of assessing bone volume? e.g. the most well-known one, i.e. thresholding? The process of manually generating a polygonal curve is very long and does not always involve an accurate determination of the volume of the object. 

Response 4: Thank you for the above question. We did not employ a semi-automatic segmentation method for assessment of bone regeneration as our study model did not include the creation of a bone defect. The materials used in our study were in the form of a sheet and bone formed on the outer side was considered as newly regenerated bone; only the volume of this bone was taken into consideration. From our literature search, we found that thresholding method was more applicable in cases where bone regeneration relative to a defect was studied. Therefore, we used Micro-Computed Tomography (Micro-CT) images to trace the new bone volume. All study analyses including volume determination of new bone was carried out by three individual researchers, and the average values are represented as the result; to the best of our knowledge, we state that the results of our study contain very minimal error.

Reviewer 3 Report

Comments and Suggestions for Authors

Review of the manuscript "Evaluation of hard and soft tissue responses to four different generation bioresorbable materials in Maxillofacial Surgery: An in-vivo animal study":

Title and abstract

- For clarity, it should be mentioned what the 4 generations of biomaterials for osteosynthesis are.

- In my opinion, the conclusions extrapolated to the treatment of the human facial skeleton are not supported by the results of the study, because it was an animal study. You should limit yourself to the conclusions of the study or soften the tone of the current ones, writing, for example, that "resorbable materials of a given generation are worth considering in the midface area." This comment also applies to the rest of the manuscript.

Introduction

- I have no comments on the content of this section. Consider dividing into subsections, e.g. Background, Rationale, Aim.

Materials and Methods

- I have no comments on this section.

Results

- I have no comments on this section.

Discussion and conclusions

- I suggest moving some of the content of the Conclusions to the Discussion. In my opinion, Conclusions should be no longer than a concise answer to the research questions.

Back matter

- I have no comments on this section.

Overall, this is a very good report from a well-conducted study. I wish the authors to overcome the problems mentioned in the Limitations section in future research.

Author Response

Comments 1: Review of the manuscript "Evaluation of hard and soft tissue responses to four different generation bioresorbable materials in Maxillofacial Surgery: An in-vivo animal study":

Title and abstract

- For clarity, it should be mentioned what the 4 generations of biomaterials for osteosynthesis are.

- In my opinion, the conclusions extrapolated to the treatment of the human facial skeleton are not supported by the results of the study, because it was an animal study. You should limit yourself to the conclusions of the study or soften the tone of the current ones, writing, for example, that "resorbable materials of a given generation are worth considering in the midface area." This comment also applies to the rest of the manuscript.

Response 1: Thank you for the above-mentioned suggestions. We have added the names of the four generation biomaterials enlisted in our study in the Title – Line 3, 4, 5, 6, 7 and Abstract – Line 22, 23 sections.

We understand the perspective of the correction mentioned above. We have made said changes into all parts of the manuscript for better clarity and understanding in the Abstract – Line 27, 28, 29, 30, 31, and Discussion – Line 779 and Conclusion – Line 804, 806, 809, 810, 811 sections.

Comments 2: Introduction

- I have no comments on the content of this section. Consider dividing into subsections, e.g. Background, Rationale, Aim.

Response 2: Thank you for the suggestion. As mentioned, we have subdivided the Introduction into Background – Line 58, Rationale – Line 127, and Aim – Line 133 sections.

Comments 3: Materials and Methods

- I have no comments on this section.

Response 3: Thank you very much for the comment.

Comments 4: Results

- I have no comments on this section.

Response 4: Thank you very much for the comment.

Comments 5: Discussion and conclusions

- I suggest moving some of the content of the Conclusions to the Discussion. In my opinion, Conclusions should be no longer than a concise answer to the research questions.

Response 5: Thank you for the above suggestion. We have shortened the Conclusion section and have added points that do not answer the research question at the end of the Discussion section – Line 778.

Comments 6: Back matter

- I have no comments on this section.

Overall, this is a very good report from a well-conducted study. I wish the authors to overcome the problems mentioned in the Limitations section in future research.

Response 6: We thank you for the comment. We are making efforts to modify the methodology in our current and future research, to incorporate more criteria and overcome the limitations mentioned in our current study.  

Reviewer 4 Report

Comments and Suggestions for Authors

Title:    Evaluation of hard and soft tissue responses to four different generation bioresorbable materials in Maxillofacial Surgery: An in-vivo animal study

Abstract: Okay.

Keywords:  Okay.

Introduction: Appropriate. 
Line: 63-65: please add some references. 
Line 98: Abbreviations should be explained when first stated.

Material& methods: 

Bone calvarian defect models generally accepted animal model for detecting newly bone formation.  Is there any specific reason for choosing hemiclips procedure? If ıt is superior to defect model, can you cite it with a reference? 

Why did you prefer square shape scaffold instead of rounded shape scaffold?

How does ImageJ software detect amount of new bone? Can you add a citation? 

Line 279: If new bone formation can be measured with imageJ, why didn't you also measure the degradation?

How did you decide the number of samples for each group? Did you do any pilot study prior to the main hypothesis?

The first planned specimen analysis was in 2nd week. Why did you choose this time point instead of 4th week? 

Discussion: 
Appropriate

Conclusion: Appropriate. 
References: Up to date. 

According to my review notes;

The article has been evaluated in general terms.  It is technically acceptable but needs minor revision. I appreciate the efforts of the authors.
With My Kind Regards.

Author Response

Comments 1: Title: Evaluation of hard and soft tissue responses to four different generation bioresorbable materials in Maxillofacial Surgery: An in-vivo animal study

Abstract: Okay.

Keywords:  Okay.

Response 1: Thank you for your comment.

Comments 2: Introduction: Appropriate. 

Line: 63-65: please add some references. 

Line 98: Abbreviations should be explained when first stated.

Response 2: Thank you for the suggestion. We have added references [11] and [12] in lines 69-70 as per the reviewer’s mention. We would like to clarify that, the full form of abbreviation in Line 104 (u-HA) has been stated in Line 72 of the Introduction - Background subsection, in the paragraph where the classification of four-generations of biodegradable materials has been described.

Comments 3: Material& methods: 

Bone calvarian defect models generally accepted animal model for detecting newly bone formation.  Is there any specific reason for choosing hemiclips procedure? If ıt is superior to defect model, can you cite it with a reference? 

Response 3: We thank you for your queries. Our team has researched bone regeneration earlier, wherein a defect model was created to assess new bone formation. We did not create a bone defect model as our study focused on elucidating the hard and soft tissue response to the implanted biodegradable material sheets, and sole assessment of the material sheets in inducing bone regeneration was not the main criteria. When using bone defect models, we generally prefer performing assessments on the maxillofacial skeleton rather than the calvarial skeleton, owing to site-specific variations of outcome.

We chose the hemoclip to hold the material sheet in the desired location, i.e., in contact with the buccal surface of rat mandible. We found placement of the hemoclip onto the rat mandible relatively simple as well as effective in preventing dislodgement of the sheet into the surrounding soft tissues. We have added this point to Line 200 of the Animal Protocol: Surgical Procedure and Sacrifice subsection of the Materials and Methods section.

The methodology was designed to assess ectopic bone formation on the outer side of the sheet to gauge the effectiveness of the material components in terms of their osteoconductive behavior. We would like to clarify that we do not consider our technique superior to that of the creation of a bone defect model.

Comments 4: Why did you prefer square shape scaffold instead of rounded shape scaffold?

Response 4: We thank you for the query. The biomaterial sheets originally generated for our research purpose were larger and had to be sized down to enable placement onto the rat mandible. We decided to implant a square-shaped sheet (5 mm × 5 mm) to preserve uniformity of the size of the sheet and to standardize the shape of the sheet across all groups, which may have not be possible in case the sheet was made into a round shape. This point has been added to Materials used subsection – Line 147 of Materials and Methods.

Comments 5: How does ImageJ software detect amount of new bone? Can you add a citation? 

Response 5: We thank you for the question. We did not use any ImageJ tool to detect the amount of new bone. Upon viewing the Sagittal aspect of Micro-Computed Tomography (Micro-CT) images of the rat mandible, it is possible to visually differentiate the host bone, material sheet location, hemoclip, and the new bone on the outer side of the sheet. After manual identification of the new bone, we used the ‘Polygonal selection tool’ of ImageJ to mark the new bone on the outer sheet surface and saved the volume of the traced area to ‘Region of Interest (ROI) Manager’. The above procedure was repeated until new bone was identified and traced in every slide. The result of all the volumes of the ROI areas was obtained from utilizing the ‘Measure’ option of ROI Manager and a cumulative value was drawn. In this manner, the average values from each study group were taken and the total volume of new bone was calculated using the formula: V = åSi × d, where V is the volume in mm3; Si, new bone area in a single micro-CT slice in mm2; and d, thickness of one micro-CT slice (0.02 mm). In this manner, only the volume of the selected area was calculated by ImageJ software.  A more detailed explanation has been added to the Assessment of New Bone Formation using Micro-CT subsection - Line 229, 235 under Materials and Methods section. A citation from our previous publication has been provided for reference [25].

Comments 6: Line 279: If new bone formation can be measured with imageJ, why didn't you also measure the degradation?

Response 6: We thank you for the question. Identification of new bone visually is feasible when Micro-CT images of the rat mandible are viewed with ImageJ software. It is not possible to measure the degradation potential of the biomaterial sheets due to the following reasons: (a) only third generation sheet is visible in any radiograph owing to higher concentration of uncalcined/ unsintered Hydroxyapatite (u-HA) component; the composition of the other biomaterial sheets renders them invisible radiographically, and (b) to estimate the accurate amount of molecular weight loss that has occurred at the said time point, it is imperative that the material sheet be extracted from the host tissue and assessed using the gel chromatography instrument.  

Comments 7: How did you decide the number of samples for each group? Did you do any pilot study prior to the main hypothesis?

Response 7: We thank you for the query. We did not perform pilot study prior to the main hypothesis. The sample size for the current study was decided based on minimal amount of specimen data required for performing statistical analyses. We also aimed to reduce the number of animals used in the study as per the Replacement, Reduction, Refinement (RRR) principle. We have added the RRR principle point to the Limitation subsection – Line 797 under the Discussion section.

Comments 8: The first planned specimen analysis was in 2nd week. Why did you choose this time point instead of 4th week? 

Response 8: We thank you for the question. It is well established that host tissue reaction to any implanted biomaterial is initiated within a few hours and may persist for days in case of acute inflammation and several weeks in case of chronic inflammation. Peak cell recruitment to initiate osteoid formation ensues around 2 weeks after injury, and hence we chose week 2 as an time point to evaluate the inflammatory reaction as well as bone deposition.

Comments 9: Discussion: Appropriate

Conclusion: Appropriate. 

References: Up to date. 

Response 9: We thank you for the comment.

Comments 10: According to my review notes;

The article has been evaluated in general terms.  It is technically acceptable but needs minor revision. I appreciate the efforts of the authors.

Response 10: We thank the reviewer for their kind comment.

Reviewer 5 Report

Comments and Suggestions for Authors

1. What is the reason for the choice of hydroxyapatite concentration for materials of the third and fourth generation? For the third generation material – 10%, for the fourth generation material – 40%. For the fourth generation material, 2 parameters change – the concentration of hydroxyapatite and the introduction of PGA.

2. How are the materials prepared? Based on the results of SEM, it is difficult to estimate the size of hydroxyapatite particles.

3. Fig. 7 – The authors are recommended to indicate in the photos the areas of formation of new bone tissue. Based on the results in the photos, it is quite difficult to assess where more or less new bone tissue has formed. Please write in more detail how the quantitative assessment of the volume of newly formed bone tissue was carried out according to computed tomography. How does the program separate new bone tissue from hydroxyapatite, which is part of the scaffold?

4. Fig. 16 – what is the reason for the decrease in the degradation rate of the fourth generation sample compared to the second generation sample, despite the presence of PGA in both types of material?

5. Line 562-566 "The bioactive/osteoconductive tendency exhibited by u-HA in third- and fourth-generation materials is because of a layer of calcium phosphate (CaP) that leaks out of the material and the concomitant fibrous tissue that is deposited around the biomaterial [29] following the initiation of degradation. The CaP layer eventually attracts and reacts with the serum protein fibronectin. " What causes the sorption of fibronectin on hydroxyapatite?

6. Line 603-605 "The carboxylate form of OCN binds freely to calcium ions and hydroxyapatite, exhibiting its role in bone mineralization, and promotes better bone quality and quantity" How does good binding to calcium ions contribute to higher synthesis by OCN cells?

7. Line 667-669 "Degradation of these materials follows a particular 667 pattern: at first, there is loss of molecular weight, followed by a reduction in strength re-668 duction, and finally, a decrease in mass." - Is this data from the authors' own research or literature data?

8. What role does hydroxyapatite play in the formation of new bone tissue?

Author Response

Comments 1: What is the reason for the choice of hydroxyapatite concentration for materials of the third and fourth generation? For the third generation material – 10%, for the fourth generation material – 40%. For the fourth generation material, 2 parameters change – the concentration of hydroxyapatite and the introduction of PGA.

Response 1: We thank you for the question. We would like to kindly mention that the third generation material sheet contains 40% uncalcined/unsintered Hydroxyapatite (u-HA), and the fourth generation material sheet contains the reduced percentage – 10% of u-HA. During our previous research analyses, we noticed that 40% u-HA concentration induces more new bone formation in a rat mandibular critical defect model. Third generation material combines u-HA along with Poly-l-lactic acid (PLLA), a first generation material. In-vivo testing of PLLA in animals has shown many disadvantages; the main factors being prolonged resorption period and induction of chronic inflammatory host response. Clinical usage of third generation material, u-HA/PLLA is also found to have similar effects. We used a decreased amount of u-HA in the fourth generation material to study whether a lesser amount of u-HA can still be adequately beneficial toward promoting bone regeneration. We attempted to decrease the overall resorption time of the material by combining u-HA with second generation material, Poly-l-lactic acid/ Polyglycolic acid (PLLA/PGA). PLLA/PGA has a faster reportion time while retaining adequate strength for the required period and this is advantageous in releasing more amount of u-HA into the vicinity, that can enable faster bone regeneration as evidenced by the results of our study. The above-mentioned justification has been highlighted under Biomaterial Induced Bone Formation - Line 584 subsection under the Discussion section.

Comments 2: How are the materials prepared? Based on the results of SEM, it is difficult to estimate the size of hydroxyapatite particles.

Response 2: We thank you for this question. The materials used in the study are commercially available and we did not fabricate them. Synthesis of u-HA was done by hydrolysis of pure calcium hydrogen phosphate anhydride and calcium carbonate heated to 90 degrees; the particles were then filtered, incubated, and dried. This generates large u-HA particles that were then crushed and sieved to obtain an average of 3-5 mm (reference provided in manuscript text). The materials were kindly provided for research purpose by Teijin Medical Technologies Co., Ltd., Osaka, Japan. The mean size of u-HA particles in both third generation sheet u-HA/PLLA and fourth generation sheet u-HA/PLLA/PGA was 3-5 mm. This has been highlighted in the Materials Used subsection – Line 152 under Materials and Methods section.

Comments 3: Fig. 7 – The authors are recommended to indicate in the photos the areas of formation of new bone tissue. Based on the results in the photos, it is quite difficult to assess where more or less new bone tissue has formed. Please write in more detail how the quantitative assessment of the volume of newly formed bone tissue was carried out according to computed tomography. How does the program separate new bone tissue from hydroxyapatite, which is part of the scaffold?

Response 3: Thank you for your suggestion. The images in Fig. 7 have been represented with yellow arrows accordingly to make new bone formation more discernible – Line 339, 340. Upon viewing the Sagittal aspect of Micro-Computed Tomography (Micro-CT) images of the rat mandible, it is possible to visually differentiate the host bone, material sheet location, hemoclip, and the new bone on the outer side of the sheet. After manual identification of the new bone, we used the ‘Polygonal selection tool’ of ImageJ to mark the new bone on the outer sheet surface and saved the volume of the traced area to ‘Region of Interest (ROI) Manager’. The above procedure was repeated until new bone was identified and traced in every slide. The result of all the volumes of the ROI areas was obtained from utilizing the ‘Measure’ option of ROI Manager and a cumulative value was drawn. In this manner, the average values from each study group were taken and the total volume of new bone was calculated using the formula: V = åSi × d, where V is the volume in mm3; Si, new bone area in a single micro-CT slice in mm2; and d, thickness of one micro-CT slice (0.02 mm). We identified the structures separately as explained in Fig. 5 and performed the marking and tracing manually. We did not use any tool in ImageJ software to automatically identify new bone and carried out all the analyses manually. As per the reviewer’s suggestion, we have refined the explanation of the methodology in Assessment of New Bone Formation using Micro-CT subsection - Line 229, 235 under Materials and Methods section in detail to enable easier understanding.

Comments 4: Fig. 16 – what is the reason for the decrease in the degradation rate of the fourth generation sample compared to the second generation sample, despite the presence of PGA in both types of material?

Response 4: We thank you for the question. Though both second and fourth generation materials contain PGA, the fourth generation contains a lesser amount of PGA (12%) than compared to second generation (15%). Fourth generation is also composed of u-HA, that could possibly contribute to the delay in degradation, as u-HA is eliminated by phagocytosis. The above point has been added to Bioresorbable Sheet Resorption Rate subsection – Line 546 under Results section.

Comments 5: Line 562-566 "The bioactive/osteoconductive tendency exhibited by u-HA in third- and fourth-generation materials is because of a layer of calcium phosphate (CaP) that leaks out of the material and the concomitant fibrous tissue that is deposited around the biomaterial [29] following the initiation of degradation. The CaP layer eventually attracts and reacts with the serum protein fibronectin. " What causes the sorption of fibronectin on hydroxyapatite?

Response 5: We thank you for your query. It has been noted that proteins from fluids surrounding biomaterials are involved in ionic exchange. Serum proteins are adsorbed by Calcium phosphate when dissolution of the biomaterial occurs. The interaction and adsorption of these factors is largely dependent on factors such as crystallinity, texture, and composition of the biomaterial. When the said proteins adsorb onto the calcium phosphate substrate, their charge, availability of free calcium, and their concentration determine further surface-coverage. These proteins then influence the formation of new calcium phosphate crystals over time. The quality, nature, and conformation of these proteins that interact on the biomaterial surface then control the cellular activity. The above explanation has been added to the Biomaterial Induced Bone Formation - Line 604 subsection under the Discussion section and a reference [37] has been provided.

Comments 6: Line 603-605 "The carboxylate form of OCN binds freely to calcium ions and hydroxyapatite, exhibiting its role in bone mineralization, and promotes better bone quality and quantity" How does good binding to calcium ions contribute to higher synthesis by OCN cells?

Response 6: Thank you for the above question. Mature, carboxylate form of Osteocalcin (OCN) is present intracellularly for secretion into bone matrix. OCN is primarily known to bind to the calcium in HA, when released into bone micro-environment by undergoing morphological change. Through the tight complex formed between OCN-HA and collagen, OCN aids in bridging the matrix and mineral components of bony tissue. This point has been incorporated in the IHC Analysis – Runx2, OCN biomarker relevance subsection – Line 648 of the Discussion section and a reference [45] has been added.

Comments 7: Line 667-669 "Degradation of these materials follows a particular 667 pattern: at first, there is loss of molecular weight, followed by a 668 reduction in strength reduction, and finally, a decrease in mass." - Is this data from the authors' own research or literature data?

Response 7: Thank you for the above question. This data is from our own observation from previous study of bioresorbable materials in-vivo that has been correlated to the existing literature. This has been mentioned in the Bioresorbable Sheet Resorption Characteristics subsection – Line 737 under the Discussion section.

Comments 8: What role does hydroxyapatite play in the formation of new bone tissue?

Response 8: We thank you for the query. This point has been explained in Biomaterial Induced Bone Formation - Line 596 subsection under the Discussion section.

Round 2

Reviewer 5 Report

Comments and Suggestions for Authors

This manuscript may be accepted for publication.